# Effectiveness of structured interventional strategy for middle-aged adolescence (SISMA-PA) for preventing atherosclerotic risk factors—A study protocol

Gomathi Munusamy[1]*, Ramesh Shanmugam[2]

1 Department of Community Heal Nursing, Centre of Postgraduate Studies, Faculty of Nursing, Lincoln University College, Selangor, Malaysia, 2 Department of Medical Surgical Nursing, Centre of Postgraduate Studies, Faculty of Nursing, Lincoln University College, Selangor, Malaysia

* gomathilingeswaran2678@gmail.com

## Abstract

### Background

Atherosclerotic risk starts at an early stage in adolescents and interventions on adolescents' lifestyles are most important. The pandemic of obesity-related morbidities like atherosclerosis among young adults and atherosclerotic risk factors for middle-aged adolescents in India is known. Yet, there is a lack of cost-effective and integrated intervention programs to treat this important health problem.

### Objective

The main objective of this study is to evaluate the effect of a 12-week structured interventional strategy program, containing integrated education and supervised physical activity interventions for middle-aged adolescents.

### Methods/design

This will be a school-based pre-experimental one-group pre-post time-series research design. The sample size is estimated to include 154 adolescents of 10th to 12th grade; aged 15–17 years will be grouped as a single arm. Data will be collected from July 2021 to January 2022. The intervention duration will be 3 months. The following measures will be evaluated before, during, and after intervention: knowledge, body mass index, level of physical activity, dietary habits, and sedentary activity.

### Discussion

We believe that the structured interventional strategy approach which includes education related to atherosclerosis, physical activity, dietary habits, and sedentary activity, and cost-effective physical activity training will be more effective in preventing atherosclerotic-related changes among middle-aged adolescents. Further, this kind of approach may be applied in similar study areas elsewhere in India.

**Data Availability Statement:** No datasets were generated or analysed during the current study. All

relevant data from this study will be made available upon study completion.

**Funding:** The authors received no specific funding for this work.

**Competing interests:** The authors have declared that no competing interests exist.

## Trial registration

Clinical Trials Registry—India (Registered Number: CTRI/2021/03/032271).

## Introduction

### Background and rationale

Atherosclerosis is a systemic and leading vascular disease worldwide [1], in which lipid, or fatty deposits, plaque, inflammatory cells, and scar tissue build up within the walls of small and large arteries supply to a variety of end-organs, it includes mainly the heart, brain, kidneys, and extremities [2]. Globally the most leading risk attributable deaths due to ischemic heart disease (IHD) for about 8.38 million and 47.8 million disability-adjusted life years (DALYs) attributed to ischemic stroke [3]. In India, 45% of deaths accounted due to coronary heart disease, stroke, and hypertension [4]. It is one of the major public health issues in India which increases the burden of morbidity and premature mortality [5].

Atherosclerosis can be linked with many lifestyle risk factors such as insufficient physical activity, overweight, obesity, high body mass index (BMI) [6], sedentary behavior (SB) [7], unhealthy dietary habits [8, 9], smoking [10, 11], blood pressure [12, 13], socioeconomic status [14], and it includes non-modifiable factors like age, gender and family history. Such factors which are posing the adolescence into major risks in later life [6]. Worldwide trends in the prevalence of obesity in children and adolescents aged 5 to 19 years showed that 5.6% in girls and 7.8% in boys [15]. Studies reported that adolescents with high BMI are associated with an increased risk of atherosclerosis in adulthood [16, 17]. A global estimate shows that 3.2 million deaths each year are associated with physical inactivity and it is one of the fourth leading risk factors for mortality. Prevalence of insufficient physical activity among adolescents aged 11–17 years in India were 71.6% girls and 69.6% boys. An adolescent should engage every day at least 60 minutes of moderate to vigorous-intensity physical activity [18], but 81% of adolescents aged 11 to 17 years do not meet WHO (World Health Organization) recommended physical activity (PA) guidelines [19].

The studies corroborated the increasing prevalence of cardiovascular risk factors, burden among adolescents [8, 20–22], and increase subclinical atherosclerosis in young adulthood [23]. Recent studies reported a marked reduction in physical exercise, poor dietary habits, and increased SB among adolescents, and the need for different intervention strategies [8, 24, 25]. Health-related education is reported to be an effective means of curbing the burden of cardiovascular diseases (CVDs) among adolescence and it is the most vital part for the prevention of atherosclerosis [26].

There is no place better than schools to initiate primordial prevention-related interventions to improve the health of adolescents and future adults. The Rashtriya Kishor Swasthya Karyakram is a part of the National Adolescent Health Programme in India, is to promotes behavioral changes among the Indian adolescents' population to prevent the recent pandemic of non-communicable diseases (NCDs) like hypertension, cardiovascular diseases, and diabetes [27]. The Sustained Development Goals (SDGs) 2030 agenda "Leaving no one behind", target-3 emphasizes to ensure healthy lives and promote well-being for all ages, a 25% relative reduction in risk of premature mortality from NCDs and 10% relative reduction in the prevalence of insufficient physical activity [28].

The Structured Interventional Strategy for Middle-aged Adolescence for Preventing Atherosclerotic risks (SISMA-PA) is intended to develop a structural interventional program and

evaluate its effectiveness, which will be suitable for the Indian adolescence and we strongly believe that this Structured Interventional Strategy (SIS) will be effective in improving PA, reducing BMI, healthy eating habits, decreasing SB, and knowledge of atherosclerotic risk factors among middle-aged adolescents.

## Methods

### Study design

The SISMA-PA study is a pre-post time-series design (Fig 1). This study was prospectively registered in CTRI (Clinical Trials Registry-India, CTRI ID: 2021/03/032271). This protocol was developed according to the SPIRIT [29] checklist (S1 File). The adolescents who are eligible for recruitment will be proportionately selected from the randomly chosen schools located in the study area and will be followed for 12 weeks. The adolescents who are overweight and obese will participate in a preventive education program on atherosclerosis, PA training, and a motivational program by providing an information booklet. The Institutional ethical review board of Narayana College of Nursing, a study center for Lincoln University College in Malaysia, examined and approved this study (Ref No. 01/PhD/LUC/2019). The formal permission obtained from the District Educational Officer, Arakonam, Tamil Nadu. The school heads will be contacted first, by telephone and visits will be scheduled. Informed written consent will be obtained from the parents or legal guardians (S2 File) and assent (S3 File) from all the enrolled middle-aged adolescents will be taken by the school health nurse.

### Participants and criteria for selection

The participants will be recruited by the school health nurse from the study area. According to the Department of Education, Ranipet district, Arakonam block, Tamil Nadu, India 2019 registry has eight higher secondary public and public aided schools [30]. There are 6284 students enrolled in grades 10 to 12. The middle-aged adolescents from the selected schools will undergo anthropometric screening (height and weight), and those who meet the Indian Academy of Pediatrics (IAP) BMI classification and are identified as overweight or obese [31] satisfying, the inclusion and exclusion criteria (Table 1) will be listed according to the school registration numbers. Simple random sampling will be used to select schools and systematic random sampling will be used proportionately to select study participants. All the children meeting the inclusion criteria will undergo clearance for maximal exercise testing and medical conditions that could affect the results of the study or cause adverse effects or limit their PA. Adolescents not satisfying this screening checklist will be referred to the community health center and excluded.

### Sample size determination

The effect of PA on BMI is the primary outcome of interest in this study. The following assumption is that pre-post intervention differences in our single group pre-post time-series design will have a small effect size (Cohen's d) of 0.30 for the level of PA from a previous pilot study performed on overweight subjects [32]. We adopted to calculate total sample size with STATA 14.0 version [33]. It was estimated that a power analysis sample size of 147 participants is needed to achieve a power of 0.95 at the alpha level of 0.05 for the repeated within-subjects and in four measurement points. With an anticipated maximum dropout of 5%, the estimated final sample required is 154 middle-aged adolescents.

| | STUDY PERIOD | | | | |
|---|---|---|---|---|---|
| | Enrolment | Post-allocation | | | |
| TIMEPOINT | $-t_1$ | Baseline $t_0$ | $t_1$ (1 to 4 weeks) | $t_2$ (5 to 8 weeks) | $t_3$ (9 to 12 weeks) |
| **ENROLMENT:** | | | | | |
| **Eligibility screen** | X | | | | |
| **Informed consent from parents** | X | | | | |
| **Assent consent** | X | | | | |
| **INTERVENTIONS:** | | | | | |
| *Structured Interventional Strategy* | | | ●━━━━━━━━━● | | |
| *Phase 1: Physical exercise 30 minutes [Warm up, Skipping, walking, and resting period]* | | | ●━━━━● | | |
| *Phase 2: Physical exercise 45 minutes [Warm up, skipping, walking, running, dancing, and resting period]* | | | | ●━━━━● | |
| *Phase 3: Physical exercise 60 minutes [Warm up, skipping, walking, running, dancing, bicycling, and resting period]* | | | | | ●━━━━● |
| *Information booklet on healthy dietary habits and reduction of screen time)* | | | ●━━━━━━━━━● | | |
| *Parents newsletter sent to home regarding physical activity, dietary habits, and screen time use* | | | ●━━━━━━━━━● | | |
| **ASSESSMENTS:** | | | | | |
| **Anthropometry:** *Height (cm), Weight (kg), Body Mass Index (BMI) (kg/m²), and Waist circumference(cm)* | X | X | X | X | X |
| *Blood Pressure (mm of Hg)* | | X | X | X | X |
| *Questionnaire to assess knowledge, level of physical activity, food habits, and sedentary activity)* | | X | X | X | X |

**Fig 1. SPIRIT 2013: Schedule of enrolment, intervention, and assessment in the SISMA–PA pre-post time-series study.** –t = timepoint for screening a study participants; $t_0$ = timepoint for baseline assessment; $t_1$ = phase1 intervention; $t_2$ = phase 2 intervention; $t_3$ = phase 3 intervention.

**Table 1. SISMA-PA: Selection criteria.**

| Inclusion criteria | Exclusion criteria |
|---|---|
| Middle-aged adolescence between 15 to 17 years | Generally sick and physically challenged |
| Gender (both) | Participants with a history of Cardiovascular abnormalities, Diabetes, Bronchial Asthma, and Chronic Obstructive Pulmonary Disease |
| Grade 10 to 12 | Not participated in a regular exercise program for past 6 months apart from school physical education period |
| Able to speak and understand regional language (Tamil) | Practicing weight management or in other physical activity trials/initiatives |
| Meets the Indian Academy of Pediatrics (IAP) criteria to be classified as overweight and obese [31] | Siblings or relatives studying in other schools will be selected for the interventional study |
| Will provide parental consent and minor assent | Use of drugs that would interfere with bodyweight |
| Willing to participate | |

## Interventions

**Description of the SISMA–PA program.** The educational intervention will impart knowledge to the study participants on atherosclerosis, related risk factors, and its prevention such as increasing levels of PA, healthy food habits, and reducing sedentary activity (Fig 2). The program will be delivered by the researcher with the aid of a liquid crystal display (LCD) presentation once a month for 60 minutes duration (3 sessions over 12 weeks). Information booklet on healthy dietary patterns and benefits of reducing SB will be discussed monthly once for 30 minutes (3 sessions over 12 weeks). Parents' newsletter will be sent monthly once for 3 months which includes strategies for reducing screen-based recreation in the home, and tips for a healthy diet, and potential consequences of excessive screen-use and physical inactivity. We strongly believe that 12 weeks of intervention will yield a 50–75% dosage response based on previous studies [34–36] of minimum clinically important difference.

**Enhanced physical activity training.** All selected study participants will undergo exercise during weekdays in three phases of intensified with low-moderate-vigorous PA (Fig 2). Each phase of the entire PA training will be designed and supervised by the physical education teacher. Adverse events, dropouts, self-reported symptoms, discomforts, health-related problems attributable to the intervention given will be recorded and reported to CTRI, will be disseminated through publication.

## Data collection and outcome measures

The primary outcomes will be the differences between pre-test (baseline), post-intervention phases concerning changes in knowledge, the effect of PA on BMI of the participants from the beginning to the end of follow-up. Post-intervention data will be scheduled at end of each session of SIS after 7 days and the level of PA will be assessed within 7 days of the last session of the exercise program. The secondary outcomes will be the differences between the intervention and changes in waist circumference (WC), blood pressure (BP), food habits, and sedentary activity in middle-aged adolescents from the baseline. A school health nurse will be a data collector, and measure anthropometric data according to WHO standardized procedures [37].

## Procedures

**Anthropometrics.** Measurements will be taken for each student in a silent location. Students will ask to wear light clothing and remove shoes before testing. Height (cm) will be measured in a full standing upright position with head straight by using a portable stadiometer

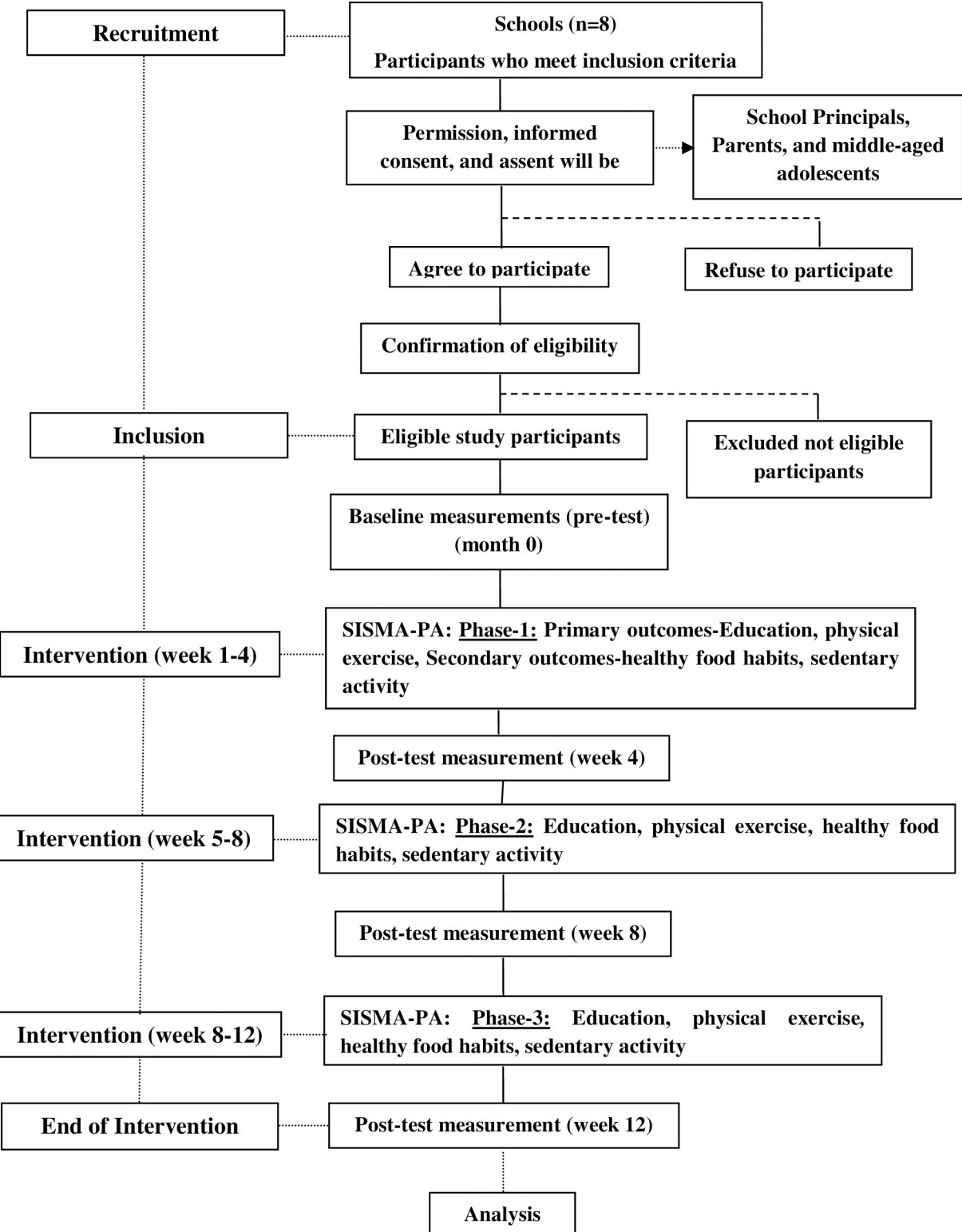

**Fig 2. SISMA-PA planned pre-post time-series study design.** SISMA-PA = Structured Interventional Strategy for Middle-aged Adolescence for Preventing Atherosclerotic risks.

with a least count of 0.1cm. Weight (kg) will be measured with the nearest 0.1kg by using a calibrated electronic weight scale. BMI will be calculated as weight (kg) divided by the height in (m$^2$). To determine the overweight/obese adolescents, the IAP reference curves using the index BMI/age, according to sex, will be used. Waist circumference measures (cm) will be taken at the nearest point of 0.1cm by placing a non-elastic tape midway between the 12[th] rib and top of the iliac crest, and at the end of normal expiration with arms in a relaxed position.

**Blood pressure.** The selected participants will be formed in small groups of 5 to a quiet location in the school. They were asked to relax and to remain silent with their feet flat on the floor, sitting upright for about 10 minutes. After 10 minutes with their left arm positioned at heart level, BP cuffs will be placed with the appropriate cuff size based on arm circumference (cm). BP will be measured using an automatic BP monitor. Three measurements will be taken with an interval of three minutes apart according to WHO guidelines. The average of the last two measurements will be taken for the analysis. BP is considered normal when the systolic and diastolic values are $< 90$[th] percentile for the adolescent's age, gender, and height [38].

**Knowledge.** A structured questionnaire (S4 File) prepared by the researcher includes general information, risk factors, symptoms, diagnosis, and prevention of atherosclerotic risk factors. The score '1' is given for the right answer and '0' will be given for the wrong answer. The total score is '65'. The composite score for knowledge will be calculated by adding the scores for all domains and rescaling them within100. Scores will then be recoded into three categories: poor score considered $\leq 50\%$; fair score considered 51–75%; good score considered 76–100%.

**Biochemical evaluation.** Blood samples will be collected for the eligible study participants after obtaining their parents/guardians' consent and assent form. Study participants will be oriented to overnight fast within 12hours. Samples will be collected by qualified professionals in the morning. Overnight blood samples will be taken for glucose, total cholesterol, triglycerides, HDL cholesterol, non-HDL cholesterol, LDL cholesterol, VLDL cholesterol, Coronary risk ratio-I, and Coronary risk ratio-II analyses.

**Physical activity.** A modified self-reported PA questionnaire will be used to measure the physical exercise performed by middle-aged adolescents. It consists of an 8-items, value from 1 to 5 for each item used in the PA composite score, simply takes the mean of these 8 items, which results in the final physical activity questionnaire for adolescents (PAQ-A) activity summary score. A score of '1' indicates low PA, whereas a score of '5' indicates high PA [39]. Then adolescents will be categorized into active who have a score of $>3$ and sedentary with a score of $<3$ [40].

**Dietary pattern.** It refers to food habits of middle-aged adolescents assessed by a modified standardized adolescent food habit checklist (AFHC). It consists of 23-items, 1 point for each 'healthy' response. The final score will be adjusted for 'not applicable' and missing responses using the formula: AFHC score = no of 'healthy' responses x (23/no of items completed) [41].

**Sedentary activity.** Total time spent in a sedentary activity like watching television/video, mobile games, working on a computer for (fun/homework), active commuting to school, and sitting or chatting with friends /phone) will be calculated for weekdays and weekends by adolescent sedentary activity questionnaire (ASAQ) [42].

## Participant adherence, retaining and end-point

The success of the SISMA-PA program strongly depends on the active participation of the participants. Participants who complete the program successfully will be rewarded with an appreciation certificate of completion and study materials worth Indian rupees 100. The attendance of the SIS and physical exercise program will be recorded during each session. Participants will

be marked as absent when unable to attend SIS one session or drops in the middle of the session and the exercise 3 sessions consecutively. The end-point of the SISMA-PA will be defined as increase knowledge on atherosclerotic risk factors >75%, completing >90% of exercise training, selecting healthy responses of food habits >90%, and decrease screen time use <2 hours/day, active commuting >20 minutes per day.

## Data management, storage, and monitoring

Data will be collected, entered, and stored on a computer into excel and SPSS data files. The participant's information will be identified only by a study number and it will be protected with a password. Access to data can be addressed to the principal investigators with an appropriate research question. Participant's drop-out will be noted and their reasons documented. We will assure the quality of data by random checking of the data entered. Data will be kept for 10 years after the research is completed and all data (electronic and hard copy) will be destroyed after the storage period. Data will be monitored by the members of IEC and CTRI.

## Data analysis

The normality of the outcome variables will be checked and the data will be expressed as mean, standard deviation (SD), or median and ranges. Recruitment, retention, and intervention fidelity will be disseminated as a percentage. A logistic regression model will be used to examine the potential predictor variables. The differences between the baseline, post-intervention, and follow-up of the continuous variables will be analyzed by Student's t-test and correlation coefficient. For the categorical variables, the chi-square test will be used. The effect of SISMA-PA for pre-post 12 weeks on the outcome variables will be expressed using analysis of variance and the principle for intention-to-treat will be applied. In addition, adherence to the interventional program will be reported in terms of the attendance percentage of each session.

## Data dissemination plan

The results of the trial will be reported according to TREND guidelines and modifications made to the protocol will be reported to CTRI. The health-related outcome of the study participants after the intervention will be informed to the parents, guardians, and the school authorities. The findings and results of the trial will be notified to CTRI and the health care professional involved in child health at Arakonam and other responsible bodies. The research findings will also be disseminated through scientific articles in peer-reviewed journals, and conferences.

## Trial status

The recruitment will commence on October 2021 and is expected to finish in March 2022.

## Discussion

The present interventional trial protocol hypothesized that SISMA-PA will impact positive change in BMI, PA, lifestyle, food habits by cost-effective intensified interventions, and that this intervention will require no special training by paid coaches, and will be enhanced by selecting low-cost seasonal fruits and vegetables; increase knowledge by theoretical based activity to promote a healthy lifestyle, and that could serve as preventive aspects in the development of atherosclerotic risk factors in adolescents. Overweight, obesity, hypertension, food consumption, lack of PA, sedentary lifestyle are important predictors for cardiovascular diseases among Indian adolescents [14, 43, 44]. The previous studies found that there is an association

between lifetime risk factors present in childhood contributes to atherosclerosis in young adulthood [9, 13, 45]. Our research will help to understand the differences and effect of continuous exercise on BMI in three phases of moderate to vigorous PA, as well as predict the individual variability in interventions among overweight/obese middle-aged adolescents.

## Supporting information

**S1 File. SPIRIT (Standard Protocol Items: Recommendations for Interventional Trials) checklist for effectiveness of SISMA-PA: A study protocol.**
(DOCX)

**S2 File. Informed consent form.**
(DOCX)

**S3 File. Assent form.**
(DOCX)

**S4 File. Questionnaire.**
(DOCX)

**S5 File.**
(PDF)

## Acknowledgments

We warmly acknowledge the District Educational Officer and school health nurse, School Heads, Arakonam taluk, Ranipet district, Tamil Nadu, India who has given support, co-operation, and permission to conduct this study.

## Author Contributions

**Conceptualization:** Gomathi Munusamy, Ramesh Shanmugam.

**Data curation:** Gomathi Munusamy.

**Formal analysis:** Gomathi Munusamy, Ramesh Shanmugam.

**Funding acquisition:** Gomathi Munusamy.

**Methodology:** Gomathi Munusamy, Ramesh Shanmugam.

**Project administration:** Gomathi Munusamy.

**Resources:** Gomathi Munusamy.

**Supervision:** Gomathi Munusamy, Ramesh Shanmugam.

**Writing – original draft:** Gomathi Munusamy.

**Writing – review & editing:** Gomathi Munusamy, Ramesh Shanmugam.

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
