## [Decision Letter · Decision Letter 0]

10 May 2022

PONE-D-22-02489Effectiveness of structured interventional strategy for middle-aged adolescence (SISMA-PA) for preventing atherosclerotic risk factors: A study protocolPLOS ONE

Dear Dr. Munusamy,

Thank you for submitting your manuscript to PLOS ONE. After careful consideration, we feel that it has merit but does not fully meet PLOS ONE’s publication criteria as it currently stands. Therefore, we invite you to submit a revised version of the manuscript that addresses the points raised during the review process.

We look forward to receiving your revised manuscript.

Kind regards,

Yoshihiro Fukumoto

Academic Editor

PLOS ONE

Journal Requirements:

Reviewers' comments:

Reviewer's Responses to Questions

**Comments to the Author**

1. Does the manuscript provide a valid rationale for the proposed study, with clearly identified and justified research questions?

Reviewer #1: Partly

Reviewer #2: Yes

2. Is the protocol technically sound and planned in a manner that will lead to a meaningful outcome and allow testing the stated hypotheses?

Reviewer #1: Partly

Reviewer #2: Yes

3. Is the methodology feasible and described in sufficient detail to allow the work to be replicable?

Reviewer #1: Yes

Reviewer #2: Yes

4. Have the authors described where all data underlying the findings will be made available when the study is complete?

Reviewer #1: No

Reviewer #2: Yes

5. Is the manuscript presented in an intelligible fashion and written in standard English?

Reviewer #1: Yes

Reviewer #2: Yes

6. Review Comments to the Author

You may also provide optional suggestions and comments to authors that they might find helpful in planning their study.

Reviewer #1: The major weakness of this protocol is that the author/s highlighted cost effectiveness in the background and it should be the main objective of the proposed study, but the author/s did not say anything of how they will calculate cost. Authors must address this issue in the data analyses.

Reviewer #2: This is a study protocol to evaluate the effect of a 12-week structured interventional strategy program, containing integrated education and supervised physical activity interventions for middle-aged adolescents. Well-written and corrdinated according to SPIRIT. I have one question.

1. Why the investigators think that 12-week is enogh o improve the awareness on prevention of atherosclerotic risk factors by structured interventional strategy, physical activity training, information on a healthy diet, and reducing sedentary activity? Usually, lifestyle-related diseases need longer intervention because of the relactance of participates against lifestyle change.

7. PLOS authors have the option to publish the peer review history of their article (what does this mean?). If published, this will include your full peer review and any attached files.

Reviewer #1: No

Reviewer #2: No

---

## [Author Response · Author response to Decision Letter 0]

1 Jun 2022

Date: May 2022

POINT-BY-POINT RESPONSE LETTER

PLOS ONE

Manuscript ID PONE-D-22-02489

Title: Effectiveness of structured interventional strategy for middle-aged adolescence (SISMA-PA) for preventing atherosclerotic risk factors: a study protocol

We express our sincere appreciation to the reviewers and editor for their time, comments, suggestions, and interest in this study protocol. The reviewers and editor had pointed out constructive comments, and legitimate questions which improved the paper.

Please find our responses below:

Note: Correction and edition made in the track change version:

 (Reviewer-1) Review questions/comments Response

1 The major weakness of this protocol is that the author/s highlighted cost-effectiveness in the background and it should be the main objective of the proposed study, but the author/s did not say anything of how they will calculate cost. Authors must address this issue in the data analyses.

 Yes, Thank you for your comments and time consideration.

We have mentioned the rationale for cost-effectiveness under the discussion section now. Page 12, Line 241-243.

Rationale: 

This intervention will require no special training by paid coaches, and will be enhanced by selecting low-cost seasonal fruits and vegetables. 

 (Reviewer-2) Review questions/comments Response

1. Why the investigators think that 12-week is enough to improve the awareness on prevention of atherosclerotic risk factors by structured interventional strategy, physical activity training, information on a healthy diet, and reducing sedentary activity? Usually, lifestyle-related diseases need longer intervention because of the relactance of participates against lifestyle change. Thank you, for your comments and time consideration.

We have mentioned the rationale under the methods section, interventions. Page 07, Line 134-135.

Rationale:

We strongly believe that 12 weeks of intervention will yield a 50-75% dosage response based on previous studies [34–36] of minimum clinically important difference.

With regards

Gomathi Munusamy

gomathilingeswaran2678@gmail.com

---

## [Decision Letter · Decision Letter 1]

4 Jul 2022

Effectiveness of structured interventional strategy for middle-aged adolescence (SISMA-PA) for preventing atherosclerotic risk factors: A study protocol

PONE-D-22-02489R1

Dear Dr. Munusamy,

We’re pleased to inform you that your manuscript has been judged scientifically suitable for publication and will be formally accepted for publication once it meets all outstanding technical requirements.

Kind regards,

Yoshihiro Fukumoto

Academic Editor

PLOS ONE

Additional Editor Comments (optional):

Reviewers' comments:

Reviewer's Responses to Questions

**Comments to the Author**

1. Does the manuscript provide a valid rationale for the proposed study, with clearly identified and justified research questions?

Reviewer #1: Yes

Reviewer #2: Yes

2. Is the protocol technically sound and planned in a manner that will lead to a meaningful outcome and allow testing the stated hypotheses?

Reviewer #1: Yes

Reviewer #2: Yes

3. Is the methodology feasible and described in sufficient detail to allow the work to be replicable?

Reviewer #1: Yes

Reviewer #2: Yes

4. Have the authors described where all data underlying the findings will be made available when the study is complete?

Reviewer #1: Yes

Reviewer #2: Yes

5. Is the manuscript presented in an intelligible fashion and written in standard English?

Reviewer #1: Yes

Reviewer #2: Yes

6. Review Comments to the Author

You may also provide optional suggestions and comments to authors that they might find helpful in planning their study.

Reviewer #1: The authors addressed all concerns in the protocol. I am happy with the author response. Congratulation !

Reviewer #2: In the revised manuscript, the authors has provided more infomation on time setting of 12-week based on previous stidies. I do not have further comments.

7. PLOS authors have the option to publish the peer review history of their article (what does this mean?). If published, this will include your full peer review and any attached files.

Reviewer #1: **Yes: **Sujarwoto Sujarwoto

Reviewer #2: No

---

## [Editor Report · Acceptance letter]

7 Jul 2022

PONE-D-22-02489R1 

Effectiveness of structured interventional strategy for middle-aged adolescence (SISMA-PA) for preventing atherosclerotic risk factors - a study protocol 

Dear Dr. Munusamy:

I'm pleased to inform you that your manuscript has been deemed suitable for publication in PLOS ONE. Congratulations! Your manuscript is now with our production department. 

Kind regards, 

on behalf of

Dr. Yoshihiro Fukumoto 

Academic Editor

PLOS ONE